# Opportunities and Challenges for Hurricane Resilience on Agricultural and Forest Land in the U.S. Southeast and Caribbean

**Sarah S. Wiener [1,\*], Nora L. Álvarez-Berríos [2] and Angela B. Lindsey [3]**

[1]   USDA Forest Service, Southeast Climate Hub, Research Triangle Park, NC 27709, USA
[2]   USDA Forest Service, International Institute of Tropical Forestry, Río Piedras, PR 00926-1119, USA; nora.l.alvarez-berrios@usda.gov
[3]   IFAS Center for Public Issues Education in Agriculture & Natural Resources, University of Florida, Gainesville, FL 32611, USA; ablindsey@ufl.edu
\*   Correspondence: sarah.s.wiener@usda.gov; Tel.: +1-281-798-2572

**Abstract:** Three storms in the 2017 hurricane season caused $265 billion in damages in the U.S. Southeast and Caribbean, including billions in losses in the agriculture and forestry sector. Climate change projections indicate that such disastrous hurricane seasons are becoming more normal. Working land management sectors need to prepare for this future. However, few studies evaluate hurricane resilience strategies, or challenges faced by land managers surrounding hurricane events. Boundary organizations are critical to hurricane preparedness and recovery, advising land managers before hurricanes, and often supporting recovery efforts. Here, we rely on public advisors' experiences to understand how land managers pursue hurricane resilience. Using focus groups and an online survey of three agencies in the Southeast U.S. and U.S. Caribbean (n = 607), we identify challenges faced by land managers before and after hurricanes, and the strategies they implement to minimize damage. We learn that land managers are faced with many diverse and unique challenges related to hurricanes, but that long-term planning for hurricane events is uncommon compared to shorter-term preparedness and recovery activities. Efforts towards hurricane resilience should incorporate local needs, align with other land management goals, and increase overall resilience to climate change and related stressors. The results of this research can guide state/territorial and national-level prioritizations regarding hurricane resilience, as well as identify research needs on hurricane resilience strategies.

**Keywords:** hurricanes; tropical storms; boundary organizations; agriculture; forestry; climate change adaptation

## 1. Introduction

The 2017 and 2018 hurricane seasons caused billions of dollars of losses in the agriculture, livestock, and forestry industries across the Southeast U.S. and U.S. Caribbean. For example, Hurricane Michael (October 2018) caused over $2 billion in losses to the timber industry in Florida, Alabama, and Georgia [1,2], while Irma (September 2017) caused $1.3 billion in damages to agriculture in Florida alone [3]. Every coastal state from Texas to Virginia and both U.S. Caribbean territories had counties with disaster designations due to hurricanes or tropical storms during these two seasons (Figure 1) [4]. These figures do not account for damages to farm structures, reduced productivity in future growing seasons, or the personal burden on land managers and their families due to trauma or economic hardship. Further, multiple studies suggest that climate change is expected to increase hurricane intensity, potentially making these storms more destructive in the future [5–8]. Despite the economic and personal hardships inflicted upon agricultural and forest land managers by hurricanes,

they remain an understudied group regarding hurricane impacts and strategies to mitigate those impacts. This dearth of information limits the ability of land managers to adapt to increasingly intense hurricanes.

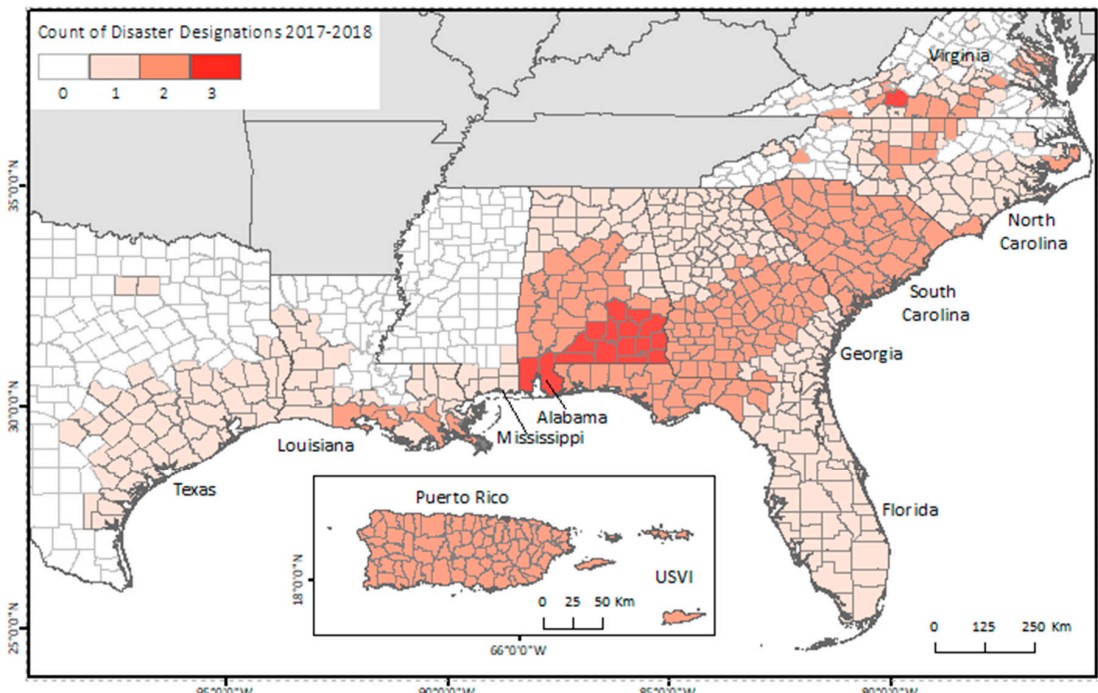

**Figure 1.** Count of presidential disaster designations in the Southeast U.S. and U.S. Caribbean due to hurricanes and tropical storms in 2017 and 2018 [4].

One group of professionals tasked with helping land managers prepare for and recover from hurricanes are land management advisors in the public sector, including employees of Cooperative Extension, the United States Department of Agriculture (USDA) Natural Resources Conservation Service (NRCS), and state forestry agencies. Given the various roles advisors play and the interactions they have with land managers before and after storms, they have valuable insight into the actions taken by land managers surrounding hurricane events. Using responses from a 2018 to 2019 survey of Cooperative Extension, NRCS, and state forestry agencies, we seek to answer three research questions: (1) What challenges do land managers face related to hurricane preparedness and recovery?; (2) what are the most important strategies land managers can implement related to hurricane preparedness and recovery?; and (3) are land managers implementing these strategies? Results from this study provide a baseline of information on hurricane preparedness and recovery in the agricultural and forestry sectors, and can inform future research and programming priorities intended to better prepare these sectors for the more intense hurricane seasons that are expected to hit the Southeast U.S. and Caribbean into the future.

### 1.1. Hurricane Impacts and Climate Change

As climate change is expected to intensify hurricanes, identifying ways to increase resilience to this level of activity becomes more critical. While the 2017 season tied the historical record for the most hurricane-strength named storms [9], this destructive season is consistent with multiple future expectations. First, though the number of hurricanes is not expected to increase, stronger storms that rapidly intensify are expected, and these storms can be more difficult to forecast, which complicates decision making regarding hurricane preparation [5,6,9]. Second, increases in heavy rainfall are also expected, as was observed during Hurricane Harvey (> 60 inches of rain), and Hurricane María, (38 inches) [7,10–12]. Third, warmer oceans can mean slower hurricane translation speeds, which

allows hurricanes more time to drop larger amounts of rainfall [8]. Though the Fourth National Climate Assessment declared that evidence is inconclusive regarding causes of slower moving hurricanes [9], a global decrease in tropical cyclone speed of 10% from 1949 to 2016 has more recently been observed and attributed to warming temperatures [8]. Ultimately, substantial evidence exists that hurricane impacts could worsen through increases in wind speed, intensification speed, and rainfall.

### 1.2. Risk Perception and Climate Change Adaptation

While evidence indicates that hurricanes are becoming more intense due to climate change, land managers may still be reluctant to implement practices that promote resilience to these events. Diverse factors can influence land managers and advisors to adopt or promote practices that are considered more resilient to climate change and extreme weather events [13], but higher perceived risk of climate and weather impacts has been repeatedly found to increase positive attitudes towards adaptation [14–18].

Additionally, the more removed, or "psychologically distant" an individual is from an event in terms of time, space, social distance, and certainty, the less influence that event and related risk perception has on decision making [19], and decisions regarding climate change adaptation tend to be driven by concerns surrounding local impacts [13,20]. For example, Haden and coauthors [15] found that farmers in California were more likely to be willing to adopt adaptation practices based on specific local climate conditions, such as water availability and summer temperatures, and Niles and coauthors [18] found that perceived weather variability had a significant influence on agricultural advisors' perception of the need for farmers to adapt.

If we apply psychological distance to hurricanes, historically infrequent return intervals could theoretically dampen risk perception. Regular return intervals of major hurricanes (category 3 or higher) in the Southeast U.S. are between 14 and 58 years [21], so land managers may be less likely to consider hurricanes as a major threat by the time a major hurricane event returns.

### 1.3. Land Management Strategies for Hurricane Resilience

While research exists on strategies to withstand hurricane related events, such as extreme precipitation, flooding, high winds, and saltwater intrusion, few studies examine the effectiveness of these strategies in the face of hurricanes, where managed land must withstand a combination of these impacts rather than a single impact. Cooperative Extension and other agencies have developed fact-sheet-type material on hurricane preparedness and recovery, but it is unclear if these resources are based on empirical evidence, and they often lack recommendations for longer term planning. Further, while multiple studies quantify the impact of hurricanes on agriculture and forestry [22–27], these studies rarely compare impact intensity based on management choices, which would provide valuable insight to land managers. Here, we review the existing literature on management strategies to reduce vulnerability to hurricanes.

In the Caribbean and Central America, evidence suggests that diversified farms and those utilizing more agroecological practices may be more resistant to hurricane damage [28–31]. Further, Holt-Giménez [28] found that impacts to Nicaraguan farms with trees and live fences were less severe, and Perotto-Baldiviezo and coauthors [32] found that landslides were less likely in Honduras where more trees were present in a system. However, trees are highly susceptible to wind damage, so if they are a cash crop (i.e., fruit and nut bearing trees) rather than a protective barrier or means for diversification, a farmer may be more vulnerable to loss [31]. Finally, one study suggests that reducing acreage of certain crops such as cotton, soybeans, and rice near the coast could limit hurricane vulnerability of the U.S. agricultural sector as a whole [23].

While these studies provide recommendations for whole-farm preparedness, evidence for commodity-specific techniques to increase hurricane resilience is needed. The majority of existing studies focus on commodities that are multi-year investments, such as pecan and citrus orchards, while annual crops are understudied. Recommendations for orchards include using specific windbreak

orientations, setting trees at the same depth in which they were raised, grafting rather than air layering trees, pruning to reduce tree size, resetting toppled trees, and irrigating and fertilizing trees after a hurricane [33–35].

Regarding forest management, more specific recommendations exist, primarily based on post-hurricane assessments in the Southeast U.S. These studies suggest a multitude of strategies to reduce vulnerability to future hurricanes such as reducing stand rotation length, diversifying stand management styles and age class of timber, and replacing loblolly pine and slash pine with the more wind-resistant longleaf pine and hardwood species [36–41]. Post-hurricane strategies include conducting prescribed burns to reduce wildfire risk, making salvage decisions based on the lean of the tree, and monitoring for pests, diseases, and invasive species [40,42].

Finally, two studies identify existing or intended adaptation actions related to hurricanes. Rodriguez-Cruz and Niles [43] found that after Hurricane María, Puerto Rican farmers reported likeliness to adopt crop diversification, integrated disease management, and acquisition of or increases in insurance plans. Second, Campbell and Beckford [44] found that, while over half of farmers in Jamaica took steps to protect their farming operation, they were more likely to take measures to protect their homes.

In order to enable hurricane resilient land management in the face of a changing climate, additional research into effective strategies is needed. Researchers can begin by testing whether strategies known to better withstand the three major impacts of hurricanes (high winds, extreme precipitation and flooding, and saltwater intrusion) are indeed effective at managing those same impacts when delivered in the form of a strong hurricane, or by incorporating comparison of management techniques when conducting damage assessments.

### 1.4. Boundary organizations

Boundary organizations are those that exist between science and decision making while maintaining accountability to both [45]. Cooperative Extension, state forestry agencies, and NRCS have been identified as boundary organizations in the land management sector that are well suited to deliver information on climate, weather, and climate change adaptation to land managers [46–49]. Employees of these agencies are also dispersed throughout their respective states in county or area offices, and frequently interact with land managers across the state through conservation planning and provision of training and technical advice. After natural disasters, these agencies are often required to put other duties on hold while they visit land managers, hold information sessions, and provide technical service. Below, we describe the three agencies involved in this study.

### 1.4.1. Cooperative Extension

The Cooperative Extension System (CES) is hosted by Land Grant Universities in each state, and provides science and assistance related to agriculture, natural resources, health, nutrition, family and consumer sciences, youth development, and more [50–52]. Extension employees provide technical assistance, training, and other services to the public, incorporating the latest scientific research. Most Extension programs have resources and trainings related to disaster preparedness and recovery, and the Extension Disaster Education Network (EDEN) is a nationwide group of Extension professionals designated with expertise in disasters.

### 1.4.2. USDA NRCS

NRCS is the USDA branch charged with providing conservation planning and technical assistance to land managers [53]. The agency designs and promotes land management conservation practices and programs. Through the Environmental Quality Incentives Program (EQIP), NRCS assists farmers with the implementation of conservation practices that address hurricane-related concerns such as soil and wind erosion, runoff, flooding, salt in surface waters, and plant health [54]. Emergency enrollment for the program is made available after natural disasters [55]. Further, the Emergency Watershed Program

provides technical and financial assistance to communities and landowners after natural disasters to reduce threats to life and property through floodplain easements and recovery activities such as the repair of conservation practices and structures (i.e., culverts, levees, and bridges) [56].

### 1.4.3. State Forestry Agencies

State forestry agencies are situated in state governments across the country, and while their exact structure may vary by state, they manage state-owned forests and are the primary providers of technical assistance to family forestland owners across the U.S. [57]. After a hurricane, these agencies provide technical assistance and advice related to evaluation of damaged trees, timber salvage, forest health (i.e., pest and disease prevention and management), tax issues, reforestation, and future management plans [58,59].

## 2. Materials and Methods

### 2.1. Online Surveys

Following Hurricanes Irma and María in 2017, we developed a survey to understand the perspectives of land management advisors from the public sector regarding hurricane preparedness and recovery across nine states (coastal states from Texas to Virginia) and two territories (Puerto Rico and the U.S. Virgin Islands). The survey was pretested with representatives from each agency (NRCS, state forestry agencies, and Cooperative Extension) and adapted based on feedback from those individuals. To distribute the survey, we contacted State Foresters, NRCS State Conservationists, and Deans or Associate Deans for Cooperative Extension to request permission to distribute the survey, as well as assistance in reaching their staff. We requested that the survey be sent to staff who assist farm and forest land managers directly and who may be involved with hurricane preparedness and recovery. From November 2018 to January 2019, requests for participation were sent three times, approximately one week apart, following the Tailored Design Method [60].

The survey included questions about the level of impact to land managers of specific hurricane-related threats, with responses ranging from no impact to devastating impact. Means ($\mu$) and standard deviations ($\sigma$) were calculated for each threat by agency and in aggregate. Subsequently, we provided a list of short-term hurricane preparedness strategies (measures taken to prepare for an existing hurricane that is forecast to make landfall in the next week or less), long-term hurricane preparedness strategies (measures taken to protect managed land from hurricanes that may come within months or years), and hurricane recovery strategies (measures taken to assess and repair damage after a hurricane). The list of strategies was gathered from Cooperative Extension, NRCS, and state forestry agency publications and fact sheets that are intended to help land managers prepare for and recover from hurricanes, as well as from conversations with experts from those same three organizations. The strategies were categorized as agriculture, livestock, forestry, or general, and respondents were only shown the general strategies and those corresponding to their self-identified primary sector of work. Respondents were asked to estimate the proportion of land managers and landowners they work with who use a strategy (hereafter referred to as "prevalence"), as well as how important each strategy is for successful hurricane preparedness/recovery (hereafter referred to as "importance"). We then identified strategies with both a high median importance and low median prevalence and labeled them as having an importance-prevalence gap. These strategies include those with a median importance of *very important* and a median prevalence of *about 50%* or lower, as well as strategies with a median importance of *important* and a median prevalence of *less than 50%* or lower.

Finally, an open-ended question about the challenges faced by land managers during hurricane recovery was analyzed for themes using Dedoose [61]. Codes were developed based on a review of the first 100 randomly selected responses. Responses were coded independently by two reviewers, with the second reviewer making a final decision when codes did not align.

### 2.2. Focus Groups

Focus groups were held at a Hurricane Resilience Workshop on November 15, 2018 in Gainesville, Florida. Attendees were primarily from Cooperative Extension and other public agencies. Most participants were from institutions in Florida, Georgia, and Alabama, but other states were represented as well. Focus groups were held to further the conversations prompted by workshop presentations on hurricanes and hurricane resilience. In total, 24 participants were divided into two groups that met twice during the workshop, resulting in four focus group sessions that were audio recorded, transcribed, and analyzed in MAX QDA Data Analysis software. Two coders met to code the first session together and to develop a codebook by which to code the other groups. Intercoder agreement was above 70%.

The survey and focus groups were approved by the University of Florida's IRB protocol #IRB201801856.

### 3. Results

### 3.1. Online Surveys

In total, 734 individuals responded to the survey. Respondents were asked if they work directly with agricultural or forest land managers as part of their job, and those who did not were excluded from the survey. The final number of respondents included in this report was 607, including 278 respondents from Cooperative Extension, 218 from USDA NRCS, and 106 from state forestry agencies. Total responses by state/territory ranged from 12 (U.S. Virgin Islands) to 144 (Mississippi). The mean number of responses by state/territory was 51 and the standard deviation was 36, indicating a bias in the geographic distribution of respondents. Respondents reported primarily working with agriculture (n = 231), followed by livestock (n = 187), and forestry (n = 174).

From the perspective of respondents, the hurricane-related effects with the greatest impact on land managers were *power outages* ($\mu$ = 3.84; $\sigma$ = 0.94), *fallen trees* ($\mu$ = 3.79; $\sigma$ = 0.94), *impassable or closed roads and transportation issues* ($\mu$ = 3.63; $\sigma$ = 0.99), and *flooding from rain* ($\mu$ = 3.56; $\sigma$ = 1.03) (Table 1). Significant differences in perceived impact between agencies were found for *flooding from rain*, *communication issues*, *gas shortages*, *lack of potable water*, and *coastal flooding/storm surge flooding*. For each effect with statistically significant differences between agencies, NRCS reported the highest impact, followed by Cooperative Extension, and State Forestry Agencies.

We subsequently analyzed the open-ended survey question, "In your opinion, what is the biggest challenge land managers/landowners you work with face during hurricane recovery?" According to respondents, the biggest challenges faced by land managers were *farm and forest management issues* (35%), which was broken down into more specific management challenges such as *timber salvage difficulties* (13%), *repair and clean up* (7%), and *infrastructure damage and repair* (7%) (Table 2). Other frequently cited challenges included those that were *environmental/ecological* (13%), *financial* (13%), and those related to seeking and receiving *government assistance and aid* (11%). Responses were further broken down by agency. On average, all agencies listed *farm/forest management* challenges as the biggest challenge, but the subsequently ranked responses differed. Respondents from Cooperative Extension listed *environmental/ecological* (16%) and *financial* (13%) challenges second and third most often, respectively. Among NRCS respondents, the second most common response was *financial* (16%), followed by *environmental/ecological* (14%). Finally, though it is technically part of *farm/forest management* challenges, *timber salvage difficulties* was the second most commonly listed challenge (40%) by state forestry agencies, followed by *markets* (14%).

**Table 1.** Responses to "Please rate the level of each impact on the land managers you work with and their ability to recover from hurricanes.".

| | NRCS (n = 200) | Cooperative Extension (n = 240) | State Forestry (n = 88) | Aggregate (n = 528) |
|---|---|---|---|---|
| Power outages | 3.87 (1.04) | 3.88 (0.88) | 3.66 (0.85) | 3.84 (0.94) |
| Fallen trees | 3.77 (0.99) | 3.75 (0.97) | 3.93 (0.70) | 3.79 (0.94) |
| Impassable or closed roads and transportation issues | 3.68 (1.01) | 3.60 (1.02) | 3.58 (0.84) | 3.63 (0.99) |
| Flooding from rain | 3.76 (1.03) [FC] | 3.53 (1.01) [F] | 3.18 (1.01) [NC] | 3.56 (1.03) |
| Communication issues (no phone or internet service) | 3.33 (1.14) [F] | 3.27 (1.11) | 2.99 (1.00) [N] | 3.24 (1.11) |
| Gas shortages | 3.14 (1.15) [F] | 3.05 (1.20) [F] | 2.68 (1.11) [NC] | 3.02 (1.18) |
| Lack of potable water | 3.02 (1.26) [F] | 2.91 (1.19) | 2.62 (1.06) [N] | 2.91 (1.20) |
| Evacuation requirements | 2.69 (1.21) | 2.54 (1.17) | 2.42 (1.04) | 2.58 (1.17) |
| Coastal flooding/storm surge flooding | 2.59 (1.63) [C] | 2.14 (1.40) [N] | 2.38 (1.36) | 2.35 (1.50) |
| Landslides | 1.88 (1.35) | 1.71 (1.13) | 1.66 (1.08) | 1.76 (1.21) |

Responses are on a 5-point Likert-type scale (1 = no impact; 2 = low impact; 3 = moderate impact; 4 = high impact; 5 = devastating impact); mean response shown with standard deviation in parenthesis. [F] Indicates significant difference ($p < 0.05$) from state forestry agency respondents, using a Kruskal-Wallis Test. [N] Indicates significant difference ($p < 0.05$) from NRCS respondents. [C] Indicates significant difference ($p < 0.05$) from Cooperative Extension respondents.

**Table 2.** Responses to the open-ended question, "In your opinion, what is the biggest challenge land managers/landowners you work with face during hurricane recovery?".

| | Total (n = 602) | Coop. Ext. (n = 278) | NRCS (n = 218) | State Forestry Agencies (n = 106) |
|---|---|---|---|---|
| **Farm/Forest Management** [1] | **35%** | **36%** | **27%** | **47%** |
| Timber salvage difficulties | 13% | 8% | 6% | 40% |
| Repair and cleanup | 7% | 8% | 6% | 8% |
| Infrastructure damage and repair | 7% | 9% | 7% | 1% |
| Labor and machinery unavailable | 5% | 5% | 4% | 7% |
| Crop and forage loss | 4% | 7% | 4% | 0% |
| Wind damage | 4% | 5% | 3% | 1% |
| Animal care and losses | 2% | 3% | 3% | 0% |
| Farm supplies unavailable (i.e., seeds, fertilizer, feed) | 1% | 2% | 1% | 1% |
| Other | 3% | 4% | 3% | 3% |
| **Environmental/Ecological** | **13%** | **16%** | **14%** | **2%** |
| Excess water | 9% | 13% | 8% | 2% |
| Water Contamination | 2% | 2% | 2% | 1% |
| Soil Degradation | 2% | 2% | 4% | 0% |
| **Financial** | **13%** | **13%** | **16%** | **9%** |
| Financial cost of recovery | 9% | 7% | 12% | 8% |
| Financial losses | 6% | 8% | 5% | 1% |
| **Government Assistance and Aid** | **11%** | **10%** | **12%** | **9%** |
| **Electricity and Fuel Shortages** | **9%** | **8%** | **11%** | **5%** |
| **Information Needs** | **4%** | **6%** | **4%** | **3%** |
| **Access and Transportation** | **4%** | **4%** | **6%** | **1%** |
| **Communication** | **4%** | **3%** | **5%** | **3%** |
| **Markets** | **3%** | **1%** | **0%** | **14%** |
| **Personal Impacts on Family and Friends** | **2%** | **4%** | **0%** | **0%** |

[1] **Bold** font indicates parent codes; normal font indicates sub codes.

Regarding perceived prevalence and importance of hurricane resilience strategies, short-term hurricane preparedness and hurricane recovery strategies ranked higher than long-term hurricane preparedness strategies (Figures 2–4). For short-term preparedness, the top five strategies rated as most important were *stock up on fuel for generators* ($\mu = 3.49$; $\sigma = 0.67$), *stock up on potable water* ($\mu = 3.39$; $\sigma = 0.79$), *inventory livestock* ($\mu = 3.36$; $\sigma = 0.77$), *move equipment from low lying areas* ($\mu = 3.28$; $\sigma = 0.80$), and *move livestock from low lying areas* ($\mu = 3.27$; $\sigma = 0.83$) (Figure 2). Among this list of 15 strategies, four had an importance-prevalence gap (strategies with a median importance of *very important* and a median prevalence of *about 50%* or lower, or strategies with a median importance of *important* and a

median prevalence of *less than 50%* or *none*): *Stock up on potable water*, *establish escape route for livestock in case of flood*, *reinforce the roofs of structures*, and *remove harvested wood from the forest*.

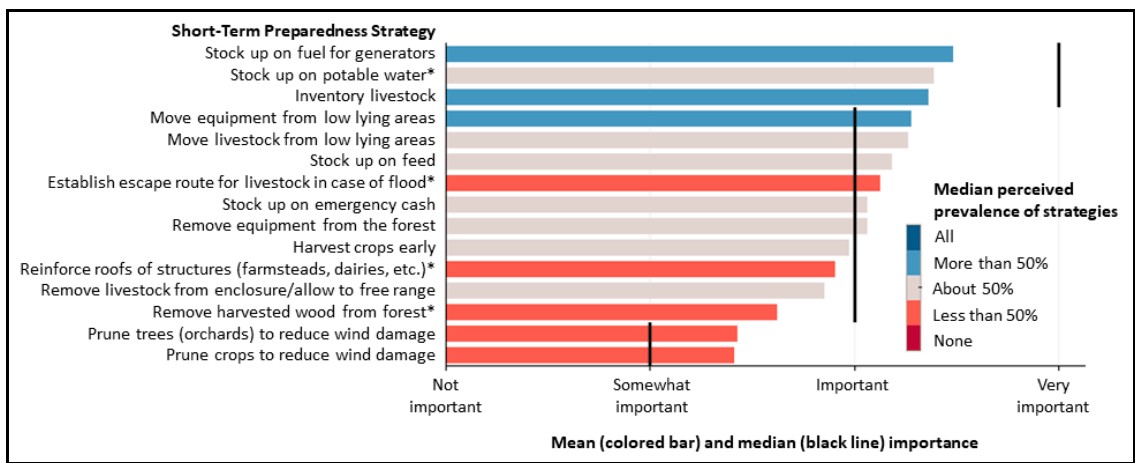

**Figure 2.** Importance and perceived prevalence of certain select short-term hurricane preparedness strategies.

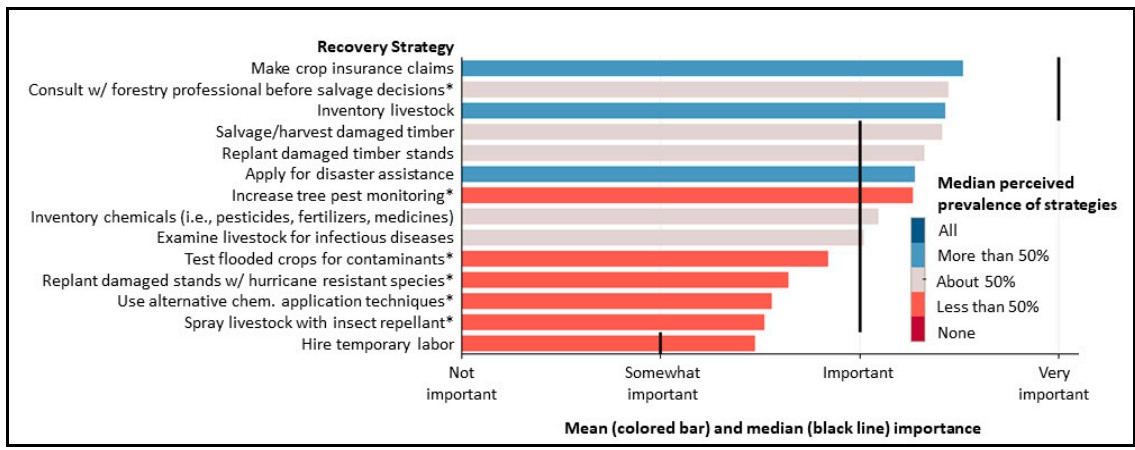

**Figure 3.** Importance and perceived prevalence of certain select hurricane recovery strategies.

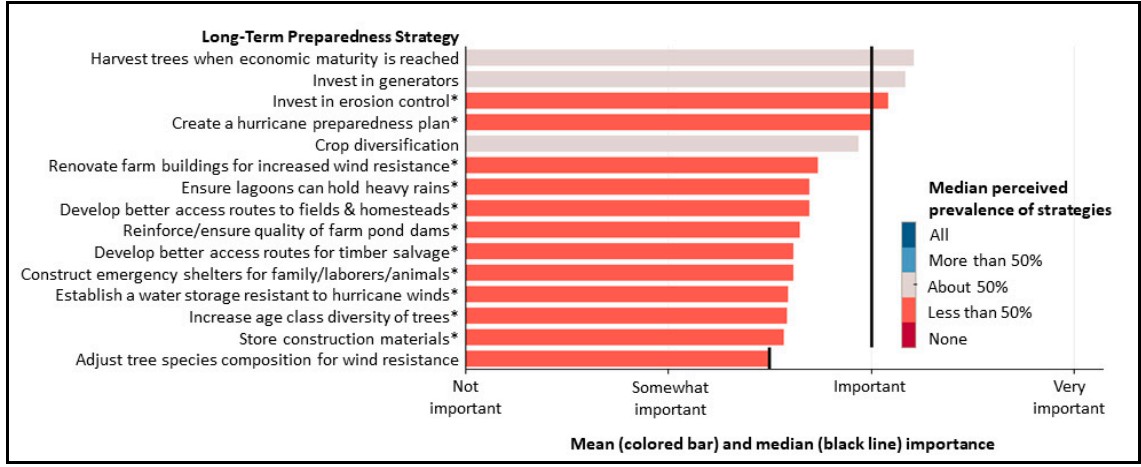

**Figure 4.** Importance and perceived prevalence of certain select long-term hurricane preparedness strategies.

Among the list of hurricane recovery steps, the top five most important strategies were *make crop insurance claims* ($\mu = 3.52$; $\sigma = 0.69$), *consult with a forestry professional* ($\mu = 3.45$; $\sigma = 0.64$), *inventory livestock* ($\mu = 3.43$; $\sigma = 0.67$), *salvage/harvest damaged timber* ($\mu = 3.42$; $\sigma = 0.58$), and *replant damaged timber stands* ($\mu = 3.33$; $\sigma = 0.62$) (Figure 3). Six of the 14 strategies had an importance prevalence gap: *Consult with a forestry professional before making decisions about salvage*, *increase tree pest monitoring*, *test flooded crops for contaminants*, *replant damaged stands with more hurricane resistant species*, *use alternative chemical application techniques*, and *spray livestock with insect repellant*.

The long-term hurricane preparedness strategies ranked as most important were *harvest trees when economic maturity is reached* ($\mu = 3.21$; $\sigma = 0.71$), *invest in generators* ($\mu = 3.17$; $\sigma = 0.81$), *invest in erosion control* ($\mu = 3.09$; $\sigma = 0.81$), *create a hurricane preparedness plan* ($\mu = 3.00$; $\sigma = 0.88$), and *crop diversification* ($\mu = 2.94$; $\sigma = 0.97$) (Figure 4). Of the 23 strategies listed, none were ranked as being used by *more than 50% of landowners*, none had a median of *very important*, and 11 had an importance–prevalence gap. This gap was greatest for *invest in erosion control* and *create a hurricane preparedness plan*.

### 3.2. Focus Groups

Focus group results generally aligned with survey results in that the most commonly discussed topics were financial impacts, operational or management impacts, and difficulties associated with preparedness and recovery efforts (Table 3).

**Table 3.** Hurricane preparedness and recovery focus group results.

|  | Percent of Code | Count of Code (N = 125) |
| --- | --- | --- |
| **Operational Impacts** [1] | **26%** | **33** |
| Land Management | 26% | 33 |
| Environmental/Ecological | 8% | 10 |
| Power/Electricity | 6% | 8 |
| Access/Transportation | 2% | 3 |
| **Preparation and Recovery Obstacles** | **36%** | **45** |
| Long-term preparation | 16% | 20 |
| Short-term preparation | 15% | 19 |
| Recovery | 14% | 18 |
| Communication & Outreach | 9% | 11 |
| **Financial Challenges** | **31%** | **39** |
| Financial Impacts | 27% | 34 |
| Government Aid | 12% | 15 |
| Market challenges | 10% | 12 |

[1] **Bold** font indicates parent codes; normal font indicates sub codes.

Participants discussed the recent impacts from Hurricane Michael and the obstacles to long-term financial recovery for the Florida Panhandle and southern Georgia. Of particular focus were commodities that become more valuable as they age, such as timber and pecans. For example, one participant said, "...in the timber industry...you can't replace [what was destroyed]. I mean [the trees] are just snapped off. There's no recovering," while another remarked that, "Pecans had driven land prices up over $8000/acre [$3237/ha]. [It was] excessive but with the money they were making they were paid that left and right and they were putting in new orchards. Now that orchard they paid $8000/acre for is worthless.".

Concern over financial resources and stability were also discussed as a barrier to preparing for the next disaster. One participant said, "[After a hurricane] my number one goal is not putting in wind breaks to think about the next hurricane, my number one goal is getting financially solvent to farm again, and that's what a lot of pecan producers and nursery people [are thinking]. It's not as much preparing for the next hurricane as it is staying in business and staying solvent.".

Further, multiple Cooperative Extension agents in the focus groups expressed frustration with the process that land managers go through to file for and receive crop insurance, particularly with

difficulties related to conducting timely damage assessments, a requirement for payments to be processed. These same agents also expressed willingness to assist in damage assessments as trusted, non-biased professionals. For example, a Cooperative Extension Agent said, "Why doesn't USDA FSA tap into us and let us do the evaluations? If they don't have enough people, we can get around our own counties, be relatively unbiased . . . and get the information . . . back to FSA to speed up processes of financial recovery for our growers, because their biggest [obstacle to recovery] is going to be financial.".

Still, participants also expressed the need for long-term and continuous planning for hurricanes, such as an Extension Agent who remarked, "My biggest takeaway [from the workshop] is preparation in general, not right before the storm, but as an annual plan that you stick to, whether it be trimming, pruning, or making sure your fuel is good...it's not right before the storm." However, participants believed that one barrier to hurricane preparedness is the lack of scientific evidence informing preparation of specific commodities for hurricanes. An Extension Agent remarked that, "[Hurricane Michael] destroyed acres and acres of pecan trees, but we don't have any information on varieties or...management [strategies]. I mean there is nothing available. The only thing we can go by is our observations...there is nothing scientific.".

Further, participants believed that the relative infrequency of hurricanes as compared to other events contributes to the lack of long-term planning. During a discussion about risk perception, a horticulturalist said, "I would imagine as you get further and further from a hurricane, [the risk perception] part of the equation becomes almost 0." Another barrier to long-term planning is that it exists in a landscape of many other challenges currently facing farm and forest owners such as debt, falling incomes, and pests and diseases. An Extension Agent commented that, "[Hurricane preparation] is an expense, so if you have other expenses [such as the tree disease] laurel wilt, you're not going to spend $300 per hour to have someone come in and prune your trees, because you're spending all your money trying to fight a disease.".

## 4. Discussion

According to the agricultural and forestry advisors in this study, short-term planning and recovery actions are critical to hurricane resilience, yet long-term planning is generally neglected by land managers. With the expected shift in hurricane intensity due to climate change, land managers and those who serve them should consider supporting long-term hurricane resilience planning as part of adaptation to climate change. While long-term planning can involve many of the strategies listed in the survey, it can also involve operational shifts and planning to enable rapid and efficient deployment of short-term preparedness and recovery efforts. For example, hurricane preparedness plans are one of the low-cost long-term strategies ranked high importance/low prevalence, and could be prioritized among advisors as part of education and outreach efforts. Development of these plans can incorporate operation-specific and crop-specific information, as well as local realities, as these factors have been found to influence adoption of resilient strategies [13]. Information on managing the highest ranked challenges, such as power outages, downed trees, and blocked roads could also be included, as could a plan for keeping livestock safe from floodwaters and other threats. Anticipating these challenges with neighbors and coordinating labor and equipment to more efficiently clear roads and manage lost power (i.e., sharing generators or fuel) could reduce the negative consequences of these impacts. Workshops held in the months following the hurricane would likely have a greater impact, as participants have less psychological distance between the hurricane events and the workshop, translating to higher likelihood to consider adoption of practices discussed at the workshop [19].

Other actions that are low cost and align with other production goals, such as resilience to more common extreme events, should be identified and prioritized as part of long-term hurricane planning. Research into practices that fall within existing cost-share programs for minimizing damage from hurricanes could inform implementation of such practices. For example, the long-term hurricane preparedness strategy with the highest prevalence-importance gap was *invest in erosion control*, which closely aligns with cost share programs under NRCS EQIP and many local Soil and Water Conservation

District programs. Strategies used to achieve this goal have co-benefits of mitigating impacts of drought and extreme precipitation, and the aforementioned programs are already tailored to local needs. These factors will likely increase the appeal of such strategies to farmers.

Other management practices shown to be more resilient to hurricanes, such as crop and stand diversification, can also reduce risk to threats such as pests, pathogens, decreasing precipitation, and increasing temperature variability [62,63]. As crop diversification was among the highest ranked long-term strategies in terms of prevalence, it may encounter less resistance among farmers. The strategies *replant damaged stands with more hurricane resistant species* (recovery), and *adjust species composition of forest stands to more wind resistant trees* (long-term preparedness) were both ranked as being implemented by *fewer than 50%* of forest landowners. Efforts to overcome barriers to adoption of more hurricane resilient strategies must involve discussions of tradeoffs between economic return and risk. For example, while evidence suggests that longleaf pine can better withstand hurricanes, as well as other threats such as drought, flooding, high winds, and beetle infestations [38,64–66], loblolly and slash pine reach maturity faster and are more economically appealing in the short-term.

Hurricane resilience can also be considered within the broader context of climate change adaptation and mitigation, from the watershed to global level. For example, practices can be prioritized to increase resilience of the watershed as a whole. Practices that increase soil-water holding capacity, such as no-till agriculture and cover cropping, can reduce the severity of flooding downstream [67], thus improving the resilience of the entire watershed to hurricanes and other extreme climatic events. Further, many of these practices, such as no-till agriculture, cover cropping, and replacing loblolly pine with longleaf pine, also have benefits of increased carbon sequestration, which contributes to mitigating the underlying causes of increased hurricane intensity [68,69]. As with all climate change mitigation and adaptation planning, local knowledge (i.e., adaptation strategies already in use by farmers and foresters) and local needs should be incorporated into such plans, involving stakeholders such as landowners and small farm and forestry organizations throughout the planning process [62]. Within this context of climate change planning, the results of this study may inform efforts to understand and improve preparedness and recovery efforts related to climate stressors in general, particularly those that are relatively infrequent yet catastrophic in nature, such as wildfires and extreme drought.

Additionally, efforts to promote more expensive investments in hurricane resilience (i.e., building more hurricane resistant structures, diversifying crop and timber systems) can be prioritized after an extreme event when rebuilding is already occurring, and as damaged and destroyed structures need replacing. Conducting outreach for long-term preparedness strategies immediately after a hurricane could increase adoption rates as the event is less psychologically distant from the decision [19,20] and rebuilding is already occurring as part of recovery.

In addition to specific preparedness and recovery actions that land managers can take, organizations that serve land managers should consider ways to enhance service provision and manage high demand for limited staff in the days, weeks, and months following a hurricane. For example, one of the highest prevalence-importance gaps was for the strategy *consult with a forestry professional before making salvage cut decisions*, while *timber salvage difficulties* and *markets* were the most commonly mentioned forestry-related challenges. A forestry professional can help determine which trees need to be harvested immediately, which should be harvested later, and recommend a forest health monitoring strategy to reduce risk of mortality, infestations, and wildfire that can occur as a result of a hurricane with up to multi-year delays [40,70]. However, availability of forestry professionals may be limited after hurricanes due to high demand and transportation difficulties. Networks of forestry professionals could consider ways to collaborate on hurricane recovery and work across boundaries to help with post-hurricane assessments. Additionally, though on-stand and in-person consultations are likely the most effective, alternative strategies could be considered to improve the reach of these professionals after a hurricane, such as an app or platform for sharing pictures of damages with professionals. An existing method for reaching land managers is to hold workshops after a hurricane where many landowners can be served simultaneously.

Among agricultural strategies, *filing for crop insurance* was ranked the most important and the most prevalent. However, qualitative data from focus groups and open-ended questions indicate that actually filing and receiving these payments is one of the larger challenges farmers face after a hurricane, and that many barriers exist to receiving these payments, such as having proper documentation and delays in official damage assessments. In most cases, these insurance or indemnity applications go through the USDA Farm Service Agency (FSA), and an FSA employee must assess or verify damages in person before a claim can be completed. However, in the wake of disasters that ravage entire counties of agricultural and forest land, the workload may be unreasonable for county staff who are likely also dealing with their own disaster recovery. Further, closed roads, difficult communication, and downed power lines could compound these issues. Cooperative Extension staff who participated in this study expressed interest in developing official partnerships that could expedite the processes of disaster payments.

Finally, the lack of empirical research into effective management strategies for hurricanes is a major limiting factor to hurricane resilience. In order to enable hurricane resilient land management in the face of a changing climate, additional research into effective strategies is needed. Researchers can begin by testing if strategies known to better withstand the three major impacts of hurricanes (high winds, extreme precipitation and flooding, and saltwater intrusion) are effective at managing those same impacts when delivered collectively in the form of a strong hurricane. This information should be delivered in easy to use formats that land managers can tailor to their unique situation, such as fact sheets and checklists.

## 5. Conclusions

The catastrophic impacts of hurricanes on agriculture and forestry in recent years cannot be ignored, especially as evidence points to increasing trends in intensity. Despite this, long-term hurricane resilience plays an undersized role in planning among land managers compared to shorter term planning and recovery efforts. Further, scientific resources to inform best practices regarding short- and long-term hurricane planning and recovery are scant for agriculture, and moderate for forestry. Though some evidence exists, more research is needed to effectively inform practice.

Boundary organizations can be key players in promoting long-term hurricane resilience through education and outreach, cost share programs, research, technical support, and more. Among boundary organizations, inter-and intra-organizational planning within and between states could lead to more efficient and effective hurricane recovery, and these agencies have an important role to play in helping land managers plan for future hurricanes. As boundary organizations encourage short- and long-term hurricane planning, they should consider practices that are location and commodity specific, aligning with other management goals, and promote resilience to multiple climate threats both on the managed land and the surrounding landscape. When possible, strategies that also sequester greenhouse gasses should be prioritized. Further, collaboration with researchers to evaluate the effectiveness of these strategies is critical to providing a base of scientific knowledge to inform future practice. As hurricanes increase in intensity, efforts to arm land managers with the tools to better withstand these storms are critical to ensuring resilient food and fiber systems, as well as rural livelihoods.

**Author Contributions:** Conceptualization, S.S.W.; Methodology, S.S.W.; Formal Analysis, S.S.W., A.B.L., N.L.Á.-B., Investigation, S.S.W., N.L.Á.-B., A.B.L., Resources, S.S.W., A.B.L., N.L.Á.-B.; Data Curation, S.S.W., N.L.Á.-B., A.B.L.; Writing—Original Draft Preparation, S.S.W., N.L.Á.-B., A.B.L.; Writing—Review & Editing, N.L.Á.-B., A.B.L., S.S.W.; Visualization, S.S.W., N.L.Á.-B., Supervision, S.S.W., Project Administration, S.S.W. All authors have read and agreed to the published version of the manuscript.

**Funding:** This research received no external funding.

**Acknowledgments:** The authors would like to thank those who pretested the survey: Sean Brogan, Libbie Johnson, Samuel Prieto Pulido, Edwin Mas, and Javier Rosario. We also thank the Cooperative Extension, NRCS, and state forestry agency employees who helped distribute and participated in the survey, as well as the focus group participants. We also thank the USDA Climate Hubs for supporting the Hurricane Workshop, where the focus groups were conducted, and the University of Florida for hosting the workshop. Finally, thank you to the two

anonymous reviewers, and our two friendly reviewers, Steve McNulty and William A. Gould, for their thoughtful feedback on the manuscript.

**Conflicts of Interest:** The authors declare no conflict of interest.

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
