# Peer review of "Opportunities and Challenges for Hurricane Resilience on Agricultural and Forest Land in the U.S. Southeast and Caribbean"

_sustainability, doi:10.3390/su12041364_

Round 1

Reviewer 1 Report

Through an online survey, challenges and strategies for hurricane resilience are identified in this study. The design of their survey is targeted (for land manages) and informative. Their analysis and conclusions are reasonable.  My only concern is that authors may explain more about how they statistically deal with different kinds of survey data, e.g., the meanings of μ and σ. As such, I suggest the acceptance of this manuscript with minor revision.

Author Response

Thank you for your thoughtful feedback and helping to strengthen this manuscript. To respond to your comment, we added text defining the symbols in the methods section (lines 199-200): “Means (μ) and standard deviations (σ) were calculated for each threat by agency and in aggregate.”

We also added text to the caption of Table 1 to indicate that the values shown are mean and standard deviation.

Reviewer 2 Report

Very nice paper, thank you for giving me the opportunity to read (and comment on) this paper.

I have minor comments since the manuscript is rather well done and can be published after minor revisions.

1) One point is about the take-home message and especially, how to incorporate local needs in climate-driven policies focusing on critical events such as hurricanes, etc.

I believe the approach proposed by 

Altieri, M. A.; & Nicholls, C. I. The adaptation and mitigation potential of traditional agriculture in a
627 changing climate. Climatic Change 2017, 140, 33-45. https://doi.org/10.1007/s10584-013-0909-y

is particularly appropriate and needs some more comments vis à vis your specific approach (the discussion seems the appopriate place to enrich this part).

2) How to generalize your empirical results to other climate events of interest for local communities (e.g. droughts, fires, hot waves, climate mixing, etc.)? A brief discussion on this point can give more value to your paper.

Author Response

Thank you for your thoughtful feedback and helping to strengthen this manuscript. Our responses are below.

Comment 1:

To your first comment, we added information on local needs and knowledge into two parts of the Discussion section, with reference to the suggested Altieri paper. As two of our authors are federal employee members of the Executive Branch, we are unable to provide policy prescriptions. Therefore, we chose to address your point within the context of planning rather than policy.

In the paragraph on considering hurricane resilience within the broader context of climate change mitigation and adaptation, we added (lines 397-401), “As with all climate change mitigation and adaptation planning, local knowledge (i.e., adaptation strategies already in use by farmers and foresters) and local needs should be incorporated into such plans, involving stakeholders such as landowners and small farm and forestry organizations throughout the planning process [Altieri & Nichols 2017].”

We also added more information on local needs to the following paragraph on lines 376-377 (underlined text is new), “For example, the long-term hurricane preparedness strategy with the highest prevalence-importance gap was invest in erosion control, which closely aligns with cost share programs under NRCS EQIP and many local Soil and Water Conservation District programs. Strategies used to achieve this goal have co-benefits of mitigating impacts of drought and extreme precipitation, and the aforementioned programs are already tailored to local needs. These factors will likely increase the appeal of such strategies to farmers.”

Comment 2:

We added to the discussion section (lines 401-404), “Within this context of climate change planning, the results of this study may inform efforts to understand and improve preparedness and recovery efforts related to climate stressors in general, particularly those that are relatively infrequent yet catastrophic in nature, such as wildfires and extreme drought.”